# ON THE OPTIMAL PRECISION OF GANS

## ABSTRACT

Generative adversarial networks (GANs) are known to face model misspecification when learning disconnected distributions. Indeed, continuous mapping from a uni-modal latent distribution to a disconnected one is impossible, so GANs necessarily generate samples outside of the support of the target distribution. In this paper, we make the connection between the performance of GANs and their latent space configuration. In particular, we raise the following question: what is the latent space partition that minimizes the measure of out-of-manifold samples? Building on a recent result of geometric measure theory, we prove a sufficient condition for GANs to be optimal when the dimension of the latent space is larger than the number of modes. In particular, we show the optimality of generators that structure their latent space as a 'simplicial cluster' - a Voronoi partition where centers are equally distant. We derive both an upper and a lower bound on the optimal precision of GANs learning disconnected manifolds. Interestingly, these two bounds have the same order of decrease: $\sqrt{\log m}$, $m$ being the number of modes. Finally, we perform several experiments to exhibit the geometry of the latent space and experimentally show that GANs have a geometry with similar properties to the theoretical one.

## 1 INTRODUCTION

GANs (Goodfellow et al., 2014), a family of deep generative models, have shown great capacities to generate photorealistic images (Karras et al., 2019). State-of-the-art models, like StyleGAN (Karras et al., 2019) or TransformerGAN (Jiang et al., 2021), show empirical benefits from relying on overparametrized networks with high-dimensional latent spaces. Besides, manipulating the latent representation of a GAN is also helpful for diverse tasks such as image editing (Shen et al., 2020; Wu et al., 2021) or unsupervised learning of image segmentation (Abdal et al., 2021). However, there is still a poor theoretical understanding of how GANs organize their latent space. We argue that this is a crucial step in better apprehending the behavior of GANs.

To better understand GANs, the setting of disconnected distributions learning is enlightening. Experimental and theoretical works (Khayatkhoei et al., 2018; Tanielian et al., 2020) have shown a fundamental limitation of GANs when dealing with such distributions. Since the distribution modeled by GANs is connected, some areas of GANs' support are necessarily mapped outside the true data distribution. When covering modes of a disconnected distribution, GANs try to minimize the measure of the generated distribution lying outside the true modes (*e.g.* the purple area on the right of Figure 1). In other words, GANs need to minimize the measure of the borders between the modes in the latent space. Considering a Gaussian latent space, minimizing this measure is closely linked to the field of Gaussian isoperimetric inequalities (Ledoux, 1996). This field aims at deriving the partitions that decompose a Gaussian space with a minimal Gaussian-weighted perimeter. We argue that the optimal partitions derived in Gaussian isoperimetric inequalities cast a light on the structure of the latent space of GANs. Most notably, a recent result (Milman and Neeman, 2022) shows that, as long as the number of components $m$ in the partition and the number of dimensions $d$ of the Gaussian space are such that $m \leq d + 1$, the optimal partition is a 'simplicial cluster': a Voronoi diagram obtained from the cells of equidistant points (see left of Figure 1 for $m = 3$ and $d = 3$).

In this paper, we apply this result to the field of GANs and show, both experimentally and theoretically, that GANs with 'simplicial cluster' latent space minimize out-of-distribution generated samples. We draw the connection between GANs and Gaussian isoperimetric inequalities by using the *precision* metric (Sajjadi et al., 2018; Kynkäänniemi et al., 2019), which quantifies the portion of generated

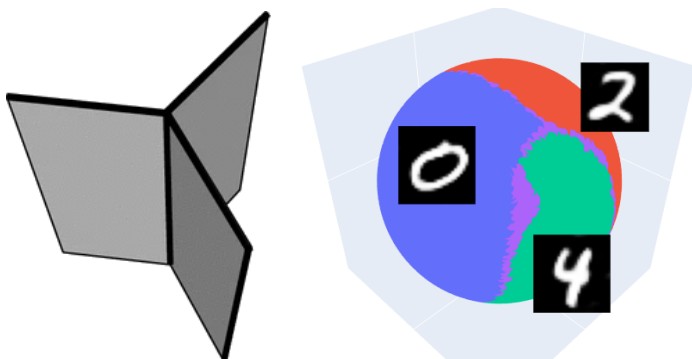

Figure 1: Illustration of the ability of GANs to find an optimal configuration in the latent space. On the left, the propeller shape is a partition of 3D Gaussian space with the smallest Gaussian-weighted perimeter (Figure from Heilman et al. (2013)). On the right, we show the 3D Gaussian latent space of a GAN trained on three classes of MNIST. Each area colored in blue, green, or red maps samples in one of the three classes. In purple, we observe the samples that are classified with low confidence. We see that the partition reached by the GAN (right) is close to optimality (left), since the latent space partition is similar to the intersection of the propeller on a sphere.

points that support the target distribution. We show that GANs with a latent space organized as a simplicial cluster reach optimal precision levels and derive both an upper and a lower bound on the precision of such GANs. Experimentally, we show that the GANs with higher performances tend to organize their latent space as simplicial clusters. To summarize, our contributions are the following:

- We are the first to import the latest results from Gaussian isoperimetric inequalities by (Milman and Neeman, 2022) to the study and understanding of GANs. We use it to show that the latent space structure has major implications on the precision of GANs.

- We derive a new theoretical analysis, stating both an upper bound and a lower bound on the precision of GANs. We show that GANs with latent space organized as a simplicial cluster have an optimal precision whose lower bound decrease in the same order as the upper bound: $\sqrt{\log m}$, where $m$ is the number of modes.

- Experimentally, we show that GANs tend to structure their latent space as 'simplicial clusters' on image datasets. First, we explore two properties of the latent space: linear separability and convexity of classes. Then, we play with the latent space dimension and highlight how it impacts the performance of GANs. Finally, we show that overparametrization helps approaching the optimal structure and improving GANs performance.

## 2 RELATED WORK

### 2.1 NOTATION

**Data.** We assume that the target distribution $\mu_\star$ is defined on Euclidean space $\mathbb{R}^D$ (potentially a high-dimensional space), equipped with the Euclidean norm $\|\cdot\|$. We denote $S_{\mu^\star}$ the support of this unknown distribution $\mu^\star$. In practice, however, we only have access to a finite collection of i.i.d. observations $X_1, \ldots, X_m$ distributed according to $\mu^\star$. Thus, for the remainder of the article, we let $\mu_m$ be the empirical measure based on $X_1, \ldots, X_m$.

**Generative model.** We consider $\mathcal{G}_L$ the set of $L$-Lipschitz continuous functions from the latent space $\mathbb{R}^d$ to the high-dimensional space $\mathbb{R}^D$. Each generator aims at producing realistic samples. The latent space distribution defined on $\mathbb{R}^d$ is supposed to be Gaussian and is noted $\gamma$. Thus, each candidate distribution is the push forward between $\gamma$ and a generator $G$ and is noted as $G\sharp\gamma$.

This Lipschitzness assumption on $\mathcal{G}_L$ is reasonable since Virmaux and Scaman (2018) has shown and presented an algorithm that upper-bounds the Lipschitz constant of any deep neural network. In practice, one can enforce the Lipschitzness of generator functions by clipping the neural networks'

parameters (Arjovsky et al., 2017), penalizing the discriminative functions' gradient (Gulrajani et al., 2017; Kodali et al., 2017; Wei et al., 2018; Zhou et al., 2019), or penalizing the spectral norms (Miyato et al., 2018). Note that some large-scale generators such as SAGAN (Zhang et al., 2019) and BigGAN (Brock et al., 2019) also make use of spectral normalization for the generator.

## 2.2 GANs AND DISCONNECTED DISTRIBUTIONS

A significant flaw of GANs (Goodfellow et al., 2014) is their difficulty in learning multi-modal distributions. This phenomenon has been analyzed by Khayatkhoei et al. (2018) and Tanielian et al. (2020). The problem comes from this fundamental trade-off: GANs can either cover all modes and generate out-of-manifold samples or generate only good quality samples and neglect some modes (mode collapse). Some methods have proposed ways to train disconnected generators (Khayatkhoei et al., 2018; Gurumurthy et al., 2017), but with little benefits compared to single overparametrized generators with rejection mechanisms (Tanielian et al., 2020; Azadi et al., 2018).

Empirically, different works give intuition on the latent space structure of GANs. Karras et al. (2019) show that binary attributes are linearly separable in the Gaussian latent space and even better separated in an intermediate latent space. Shen et al. (2020) stress that face attributes are separated by hyperplanes, and edit images only by moving in the latent space orthogonally to these hyperplanes. Arvanitidis et al. (2018) and Chen et al. (2018a) view the latent space of generative models with a Riemannian perspective and define a metric tensor using the generator's Jacobian to find the shortest paths on the data manifold.

However, these findings might not be sufficient for a clear understanding of the required geometry of the latent space. For instance, Karras et al. (2019) use a very large latent space dimension ($\mathbb{R}^{512}$), while Sauer et al. (2022) argue that the optimal latent space dimension is close to the intrinsic dimension of images ($\mathbb{R}^{64}$ for ImageNet). Tanielian et al. (2020) stress the relevance of this problem by showing that the precision of GANs can converge to 0 when the number of modes or the distance between them increases. In this paper, we make a step towards a better understanding of the behavior of GANs and expose an optimal latent space configuration when the number of modes $m$ and the dimension of the latent space $d$ are such that $m \leq d + 1$.

## 2.3 EVALUATING GANs

When learning disconnected manifolds, Sajjadi et al. (2018) illustrated the need for new measures that simultaneously evaluate the quality (Precision), and the diversity (Recall) of the generated samples. Kynkäänniemi et al. (2019) highlighted an important drawback of this PR metric: it cannot correctly interpret situations when large numbers of samples are packed together. They propose an Improved PR metric based on the non-parametric estimation of manifolds to correct this.

**Improved PR metric:** Informally, for a generator $G$, precision ($\alpha_G$) quantifies the proportion of generated samples that can be approximated with true samples, while recall ($\beta_G$) measures the proportion of true samples that can be approximated with generated ones. Applying this to GANs, using the target distribution $\mu^\star$ and modeled distribution $G\sharp\gamma$, the Improved PR metric was shown, by Tanielian et al. (2020, Theorem 1), to be asymptotically equivalent to:

$$\alpha_G^n \underset{n\to\infty}{\to} \alpha_G = G\sharp\gamma(S_{\mu^\star}) \quad \text{and} \quad \beta_G^n \underset{n\to\infty}{\to} \beta_G = \mu^\star(S_{G\sharp\gamma}), \tag{1}$$

where $S_{\mu^\star}$ denotes the support of $\mu^\star$. More recently, Naeem et al. (2020) have shown that the Improved PR metric (Kynkäänniemi et al., 2019) is not robust to outlier samples of both the target and the generated distribution. To correct this and fix the overestimation of the manifold around real outliers, Naeem et al. (2020) propose the Density/Coverage metric.

**Density/Coverage:** Instead of counting how many fake samples belong to a real sample neighborhood, density counts how many real sample neighborhoods contain a generated sample. On the other hand, coverage counts the number of real sample neighborhoods that contain at least one fake sample.

In the next section, we will use the notion of precision and recall defined in (1). Using this definition allows us to circumvent the non-parametric estimators involved in the existing metrics (Kynkäänniemi et al., 2019; Naeem et al., 2020).

## 3 Determining optimal precision in GANs

We want to better understand the latent space of GANs and stress which GANs have the highest precision under specific constraints. GANs are push-forward distributions of a unimodal (connected) Gaussian distribution $\gamma$ and a continuous function parameterized by $G$. Consequently, the modeled generative distribution $G\sharp\gamma$ will have connected support.

When learning a target distribution $\mu^\star$ with disconnected manifolds, GANs necessarily map fake data points out of the true manifold. This leads us to the following question: given that a generator samples data points in each of the distinct modes, what can be its maximum precision? To begin with, let's assume a target distribution $\mu^\star$ composed of $m$ disconnected modes.

**Assumption 1** (Disconnected manifolds). *The target distribution $\mu^\star$ lays on m equally measured spheres $M_i, i \in [1,m]$ of radius r, each located at equal distance D (with $D/2 >> r$).*

The use of Assumption 1 is reasonable. First, on many real-world datasets, data is correctly balanced in between the different modes. The equal distance assumption can be justified from the concentration of distances in high-dimensional spaces: centers of modes will be approximately at equal distance (Beyer et al., 1999; Aggarwal et al., 2001). It has also been shown that embeddings of deep neural networks trained for classification tend to collapse around means that are equidistant to one another (Kothapalli et al., 2022). This could thus pave the way for a new analysis where the chosen distance is no longer the euclidean distance in $\mathbb{R}^D$ but a distance in the feature space of the generator or any pre-trained classifier. We also further discuss how this assumption could be relaxed. For the rest of the paper, let us define the set of *well-balanced* generators that maps equally data points to the different modes of the data distribution:

**Definition 1.** *A generator G is well-balanced, if for all $i \in [1,m]$, $G\sharp\gamma(M_1) = \ldots = G\sharp\gamma(M_m)$.*

Considering well-balanced generators is also fair since many empirical improvements such as WGAN-GP (Gulrajani et al., 2017) or BigBiGAN (Donahue and Simonyan, 2019) have significantly decreased the mode collapse. GANs generate diverse output distribution on datasets such as CIFAR10, CIFAR100, and ImageNet. To validate the use of well-balanced generators, we ran a small experiment and evaluate the proportion of each class generated by GANs on MNIST and CIFAR10. On MNIST, the minimal proportion of a class is 9.2 and the maximal 10.9, while it is respectively 8.3 and 11.9 on CIFAR10 (in %). The variance/mean ratio is equal to 0.03 for MNIST and 0.22 for CIFAR10.

### 3.1 Precision and the associated partition

Now that the prerequisites for both the data and the model have been given, we propose to define our approach. We create the connection between the set of generators from $\mathbb{R}^d$ to $\mathbb{R}^D$ and the set of partitions in the latent space. In particular, for each given partition in $\mathbb{R}^d$, there exists a set of associated generators defined as follows:

**Definition 2.** *For a given partition $\mathscr{A} = \{A_1, \ldots, A_m\}$ on $\mathbb{R}^d$, we say that G is associated to $\mathscr{A}$ if:*

$$\text{for all } i \in [1,m], \text{ for all } z \in A_i, \ i = \underset{j \in [1,m]}{\arg\min} \, d(G(z), M_j), \quad \text{where } d(X, M_j) = \underset{y \in M_i}{\min} \|X - y\|.$$

It is clear that each given generator is associated with a unique partition in the latent space. Moreover, the geometry of the partition partly explains its behavior and performance. We are interested in maximizing the precision of generative models. Any point in the intersection of two cells $A_i \cap A_j, (i,j) \in [1,m]^2$ is at equal distance of $M_i$ and $M_j$ and thus does not belong to any of these modes (since $D/2 >> r$). Besides, due to the generator's Lipschitzness, there is a small neighborhood of the boundary such that any points in this neighborhood will be mapped out of the target manifold. This region in the latent space thus reduces the precision. For a given $\varepsilon > 0$, we now define the epsilon-boundary of the partition $\mathscr{A}$ as follows:

**Definition 3.** *For a given partition $\mathscr{A} = \{A_1, \ldots, A_m\}$ of $\mathbb{R}^d$ and a given $\varepsilon \in \mathbb{R}_+^\star$, we denote $\partial^\varepsilon \mathscr{A}$ the epsilon-boundary of $\mathscr{A}$, defined as follows:*

$$\partial^\varepsilon \mathscr{A} = \bigcup_{i=1}^{m} \left( \cup_{j \neq i} A_j \right)^\varepsilon \setminus \left( \cup_{j \neq i} A_j \right),$$

where $A^\varepsilon$ corresponds to the $\varepsilon$-extension of set $A$. To better understand the link between the precision of a generator $\alpha_G$ and its associated partition $\mathscr{A}$, we state the following lemma:

**Lemma 1.** *Assume that Assumption 1 is satisfied and $\mathscr{A}$ be a partition in $\mathbb{R}^d$. Then, any generator $G \in \mathscr{G}_L$ associated with $\mathscr{A}$ verifies:*

$$\alpha_G \leqslant 1 - \gamma(\partial^{\varepsilon_{min}} \mathscr{A}), \quad where \ \varepsilon_{min} = D/L. \tag{2}$$

Interestingly, this result holds independently of the partition $\mathscr{A}$. This result highlights that the geometry of the partition gives an upper-bound on the precision of the generator. Consequently, to properly determine this bound on the precision levels of generative models, one might be interested in determining the measure of this epsilon-boundary $\partial^\varepsilon \mathscr{A}$. Furthermore, to exhibit generative models with optimal precision levels, one must look at partitions with the smallest epsilon-boundary measures $\gamma(\partial^\varepsilon \mathscr{A})$. This is tightly connected to the theoretical field of Gaussian isoperimetric inequalities.

### 3.2 Optimality in GANs

Isoperimetric inequalities link the measure of sets with their perimeters. More specifically, isoperimetric inequalities highlight minimizers of the perimeter for a fixed measure, *e.g.* the sphere in an euclidean space with a given Lebesgue measure. Gaussian isoperimetric inequalities study a similar problem in Gaussian space. Borell (1975) and Sudakov and Tsirel'son (1978) show that in a finite-dimensional Gaussian space, among all sets of a given measure, half-spaces have a minimum Gaussian perimeter. More formally, for any Borel set A in $\mathbb{R}^d$ and a half-space $H$, if we have $\gamma(A) \geqslant \gamma(H)$, then $\gamma(A^\varepsilon) \geqslant \gamma(H^\varepsilon)$ for any $\varepsilon > 0$, where $A^\varepsilon$ denotes the $\varepsilon$-extension of the set $A$.

The Gaussian multi-bubble conjecture was formulated when looking for a way to partition the Gaussian space in *m* parts and with the least-weighted boundary. It was recently proved by Milman and Neeman (2022) who showed that the best way to split a Gaussian space $\mathbb{R}^d$ in *m* clusters of equal measure, with $2 \leqslant m \leqslant d + 1$, is by using 'simplicial clusters' obtained as the Voronoi cells of *m* equidistant points in $\mathbb{R}^d$. Convex geometry theory tells us that each cell is a convex cone, whose borders are hyperplanes going through the origin of $\mathbb{R}^d$. We note $\mathscr{A}^\star$ any partition corresponding to this optimal configuration, see Figure 1.

The aim of the present paper is to leverage this result to better understand the behaviour of GANs. We argue that in the case where $m \leqslant d + 1$, optimal models in levels of precision are closely linked to the optimal partitions $\mathscr{A}^\star$ derived in the Gaussian Multi-Bubble conjecture (Milman and Neeman, 2022). Besides, using results on the Gaussian boundary measure of those sets (Schechtman, 2012), we can also derive an upper-bound on the maximal precision of generative models, as follows:

**Theorem 1 (Upper-bounding the precision).** *Assume that Assumption 1 is satisfied and $m \leqslant d + 1$. For any $\delta > 0$, if L is large enough, then for any well-balanced generator $G \in \mathscr{G}_L$, we have:*

$$\alpha_G \leqslant 1 - \gamma(\partial^{\varepsilon_{min}} \mathscr{A}^\star) + \delta.$$

*In particular, there exists L with $L \geqslant D\sqrt{\log(m)}$, such that for any well-balanced generator $G \in \mathscr{G}_L$:*

$$\alpha_G \leqslant 1 - \varepsilon_{min}\sqrt{\log m}\, e^{-3/2}, \quad where \ \varepsilon_{min} = D/L. \tag{3}$$

Theorem 1 links the precision of well-balanced generators with the optimal partition from Milman and Neeman (2022). In particular, result in (3) gives an interesting insight when training GANs on a finite number of modes. Tanielian et al. (2020, Theorem 3) showed a similar result but for the asymptotic case when the number of modes increases:

$$\alpha_G \overset{m \to \infty}{\lesssim} e^{-\frac{1}{8}\varepsilon_{min}^2} e^{-\varepsilon_{min}\sqrt{\log(m)/2}} \tag{4}$$

For $\varepsilon_{min}\sqrt{\log(m)} = o(1)$ both (3) and (4) have the same behaviour w.r.t. to the number of modes. Now, to further show the usefulness of $\mathscr{A}^\star$, we prove the following theorem:

**Theorem 2 (Lower-bounding the precision).** *Assume that Assumption 1 is satisfied and $m \leqslant d + 1$. For any $\delta > 0$, there exists C large enough and $L \geqslant D\sqrt{m}\sqrt{\pi \log(Cm)}$, and a well-balanced generator $G^\star \in \mathscr{G}_L$ associated with $\mathscr{A}^\star$ such that for any other well-balanced generator $G \in \mathscr{G}_L$, we have:*

$$\alpha_{G^\star} \geqslant \alpha_G - \delta. \tag{5}$$

*Moreover, if $m \leq d$:*

$$\alpha_{G^\star} \geqslant 1 - \varepsilon_{max}\sqrt{\pi \log(Cm)} \quad where \ \varepsilon_{max} = \frac{D\sqrt{m}}{L}. \tag{6}$$

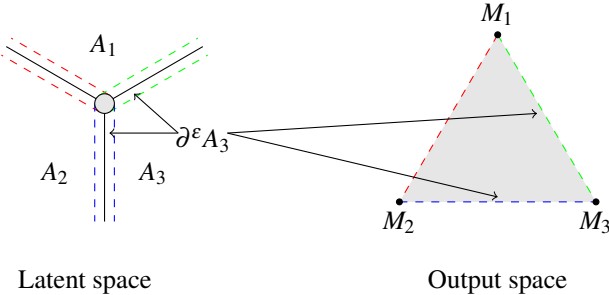

Figure 2: An optimal generator maps a 2D latent space to a 2D output space with three modes $(M_1, M_2, M_3)$. The latent space has an optimal 'simplicial cluster' geometry. In the latent space, all the $\varepsilon$-boundaries intersect each other in the gray circle, which is mapped in the output space in the convex hull of the three modes.

This theorem, which proof is delayed to Appendix, shows that the set of generators associated with $\mathscr{A}^\star$ contains optimal generators w.r.t. precision. More importantly, it shows that when $L$ is large enough, the bound in (3) may be tight, as it is almost reached by optimal generators defined in Theorem 2. An example of such optimal generators for the 2D case is given in Figure 2. This specific generator memorizes the dataset, since all samples are mapped to one of the center of the modes $M_i, i \in [1, m]$, except for those in $\varepsilon$-boundaries. It is not clear however, whether those are the only generators with optimal precision. We see that when the number of modes is less than the number of dimensions in the latent space, the only factor that impacts the precision is the number of modes.

**What if modes are not equally distant?** This assumption is needed for the definition of a well-balanced generator as the one proposed in Figure 2. In $\mathbb{R}^2$ for example, if there is no assumption at all on the location of the modes, there might not be any well-balanced generators associated with the optimal partition $\mathscr{A}^\star$. As shown in Figure 3, the latent space configuration obtained by the GANs on 3 aligned data points (right) is made of two parallel hyperplanes, much different from $\mathscr{A}^\star$ defined by Milman and Neeman (2022) (left).

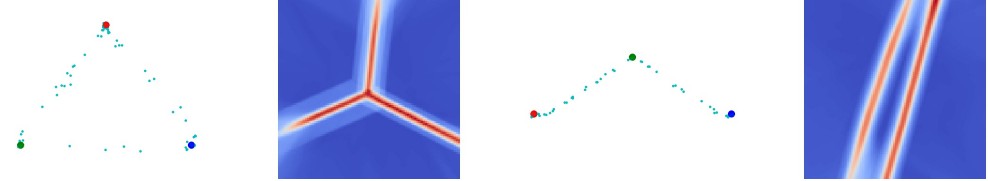

Figure 3: GANs training with 3 equidistant modes and 3 (almost) aligned modes. The first and third figure show the data points in the output space. The second and fourth stress the boundaries in the latent space using heatmaps of the norm of the gradient of the generator.

**What if the dimension $m > d + 1$?** This setting is much more complex. First, Assumption 1 is no longer valid, since one cannot find $m$ points equally distant from one another. Second, the result from Milman and Neeman (2022) is no longer valid and the optimal partition of the Gaussian space in $m$ equal cells is unknown. In this generalized context, GANs could hint at the optimal partition geometry. We show in Figure 4 examples when training a GAN from $\mathbb{R}^2$ to $\mathbb{R}^m$ with $m$ equidistant target points in $\mathbb{R}^m$. This could give some insights on how to divide the Gaussian space into $m$ equitable areas with least Gaussian-weighted perimeter.

## 4 EXPERIMENTS: UNDERSTANDING THE LATENT SPACE OF GANS

In the following experiments, we validate our theoretical analysis and derive insights for GANs trained on toy and image datasets. We verify if the latent space geometry of GANs has similar

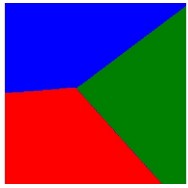 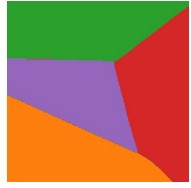 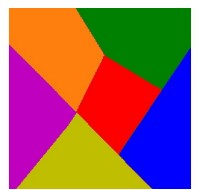 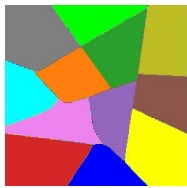

Figure 4: Extending the multi-bubble conjecture when $m > d + 1$. We plot the partition of the $\mathbb{R}^2$ latent space of a GAN that maps to $m$ equidistant points in $\mathbb{R}^m$, from $m = 3$ (left) to $m = 12$ (right). Each colored cell maps to a distinct data point in the output space.

properties than the minimizers of the Gaussian isoperimetric inequality (Milman and Neeman, 2022), and compute a series of experiments on the latent space of GANs to better understand its properties.

In all the following experiments, we train WGANs with gradient penalty (Arjovsky et al., 2017; Gulrajani et al., 2017). For mixture of Gaussians, generator and discriminator are MLP networks. For MNIST (LeCun et al., 1998), both the generator and discriminator are standard convolutional architectures. On CIFAR-10 and CIFAR-100 datasets (Krizhevsky et al., 2009), we use a Resnet-based (He et al., 2016) convolutional architecture with self-modulation in the generator (Chen et al., 2018b), and the transformer-based architecture from Jiang et al. (2021). To evaluate the performance of GANs, we use both the precision (Kynkäänniemi et al., 2019), the FID (Heusel et al., 2017), and the density/coverage (Naeem et al., 2020). As recommended by recent works (Naeem et al., 2020; Kynkäänniemi et al., 2022), we use a dataset-specific classifier to extract image features instead of an ImageNet pre-trained classifier, and thus refer to the FID as FD for Fréchet Distance. Implementation details are given in Appendix and code is provided in Supplementary Material.

## 4.1 LINEAR SEPARABILITY AND CONVEXITY

Milman and Neeman (2022) show that the optimal configuration in the latent space is obtained as the Voronoi cells of $m$ equidistant points in $\mathbb{R}^d$, if $m \le d + 1$. This means that if GANs reach this optimal configuration, each of the cells must be *convex polytopes* and thus verify the following properties: 1) each cell has 'flat' sides, and are bounded exclusively by faces; 2) each cell is convex. In the following experiments, we study whether GANs' latent spaces feature these two properties.

**Are classes linearly separable in the latent space of GANs?** To verify this, we leverage a labeled dataset and investigate if a simple linear model (*e.g.*, multinomial logistic regression) can map from latent space to label space. If cells in the latent space are bounded by hyperplanes, the linear model is expected to be a good predictor of a generated sample's label.

We consider a labeled dataset of samples with a fixed number of classes. $G_\theta$ is a pre-trained generator, and $C_\phi$ a pre-trained classifier considered as an oracle. Using $G_\theta$ and $C_\phi$, we construct a dataset of latent vectors $z \in \mathbb{R}^d$ and their associated labels $y = C_\phi(G_\theta(z))$. On CIFAR-10/100, similarly to Razavi et al. (2019), only data points with above a confidence threshold are accepted. This dataset is later split into 100k training points and 10k test points. The mapping from latent vectors $z$ to their labels $y$ is learned by a multinomial logistic regression. We report the test-set results in Table 1 under the column *LogReg Accuracy*. This accuracy reaches 90% on MNIST and 70% on CIFAR-10. Interestingly, there is also a correlation between the linear separability of the latent space and the precision metric, which validates the optimality of the simplicial cluster partition.

**Are classes convex in the latent space of GANs?** In this experiment, we draw two random latent vectors $z_0$ and $z_1$ that belong to the same class. Then, we generate linear interpolations $z_\varepsilon = \varepsilon z_0 + (1 - \varepsilon)z_1$ and verify if these new samples belong to the same class as $z_0$ and $z_1$, *i.e.* whether $C_\phi(G_\theta(z_\varepsilon))$ equals to $C_\phi(G_\theta(z_0))$. We report the mean accuracy of this experiment in Table 1 under the column *Convex Accuracy*. Again, the higher the precision, the 'more convex' each cell in the latent space seems to be. For a qualitative evaluation, we show this phenomenon in Figure 5 and stress that linear interpolations conserve the image class.

| Dataset | Architecture | Latent dim | Precision | LogReg Acc. | Convex Acc. |
|---------|-------------|-----------|-----------|-------------|-------------|
| 100 Gauss. | MLP | 100 | 75.5 | 78.5 | 87.2 |
| MNIST | CNN | 64 | 93.2 | 90.4 | 98.7 |
| CIFAR-10 | ResNet | 64 | 75.6 | 65.3 | 75.2 |
| CIFAR-10 | Transformer | 256 | 79.5 | 70.7 | 84.3 |
| CIFAR-100 | ResNet | 64 | 65.9 | 34.4 | 47.1 |
| CIFAR-100 | Transformer | 64 | 70.1 | 26.5 | 42.3 |

Table 1: In this experiment, we verify 1) if the latent spaces of GANs are linearly separable (LogReg Accuracy); 2) if each cell of the latent space is convex. (Convex Accuracy). In par with Theorem 1, the higher the precision is, the more each cell in the latent space is linearly separable and convex. The architecture *Transformer* refers to the TransGAN model from Jiang et al. (2021).

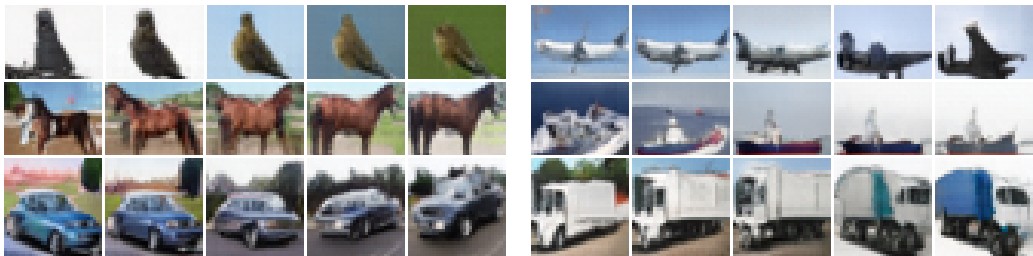

Figure 5: Illustration of convexity of classes in the latent space of GANs trained on CIFAR-10. Visual inspection confirms that latent linear interpolations between two samples of the same class most often conserve the class.

**Impact of the latent space dimension.** Now, we evaluate the impact of the latent space dimension on MNIST, CIFAR10 and CIFAR100 datasets. To do so, we vary the number of latent dimensions $d$ from 2 to 128 for each given dataset. In Figure 6, we exhibit two phases in the performance of GANs when changing the number of latent dimensions: first, when $d$ is below a dataset-specific threshold, the precision of the model falls when reducing the number of latent dimensions. When $d$ is above this threshold, the precision becomes constant, and increasing the number of latent dimensions does not bring any apparent improvement. This threshold is a function of the complexity of the dataset and its number of modes. We observe from Figure 6, that the more complex the dataset, the more it requires a large latent space for high precision levels. This is coherent with the theoretical results, where the precision decreases w.r.t. to the number of modes in $\sqrt{\log(m)}$ when $m \leqslant d + 1$.

**Disentangling the manifold dimension.** As discussed in Roth et al. (2017), two different problems can arise when training GANs: i) *dimensional misspecification* where the true and modeled distributions do not have density functions w.r.t. the same base measure, and ii) *density misspecification*, where GANs try to fit a disconnected manifold with a unimodal disitribution. To isolate the density misspecification studied in the current paper, we train a conditional GAN with a low-dimensional latent space $\mathbb{R}^d$ (*e.g.* $\mathbb{R}^5$ in our setting), so that the dimension of the generated manifold is at most 5. We later collect a dataset of synthetic generated samples *Synthetic CIFAR-10*, and train unconditional GANs by varying the dimension of the latent space. Figure 6 shows both the Synthetic CIFAR-10 and the standard CIFAR-10 converge to the same limits for FD, Precision and Density, showing that with large latent space dimensions, the *density misspecification* seems to be the main issue to cope with. A synthetic experiment showing the importance of density misspecification over dimensional misspecification is available in appendix.

**Impact of overparametrization.** Balaji et al. (2020) already showed the importance of GANs' overparametrization in both their convergence and performance. Knowing that, we study whether overparametrization can help GANs obtain the optimal geometry of latent space. In Table 2, we vary the width of ResNet generators, and highlight that overparametrized GANs better fit the target distribution. More importantly, we observe that overparametrization helps achieving better linear separability of their latent space, as shown by *LogReg Accuracy*.

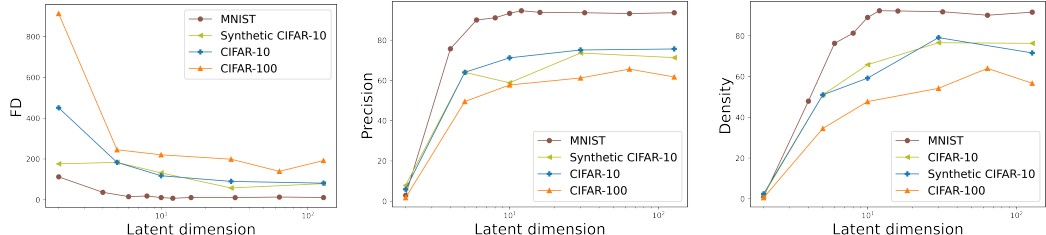

Figure 6: Performance of GANs w.r.t. the number of modes and latent space dimensions on image datasets. Left, the FD gets lower with increased latent dimension. Center and right, precision and density improve when the latent dimension increases and saturates from a threshold.

| Dataset | Width | LogReg Acc. | Convex Acc. | FD | Prec. | Dens. | Cov. |
|---|---|---|---|---|---|---|---|
| CIFAR-10 | 32 | $53.4 \pm 0.5$ | $61.1 \pm 0.3$ | $447.9 \pm 15.6$ | $63.8 \pm 0.7$ | $52.9 \pm 1.3$ | $73.0 \pm 1.0$ |
| | 64 | $60.7 \pm 0.5$ | $72.1 \pm 0.6$ | $154.3 \pm 10.8$ | $72.1 \pm 0.4$ | $69.4 \pm 1.5$ | $82.9 \pm 0.7$ |
| | 128 | $63.4 \pm 0.4$ | $73.1 \pm 0.6$ | $94.5 \pm 11.4$ | $74.9 \pm 1.0$ | $74.2 \pm 2.3$ | $85.7 \pm 1.0$ |
| | 256 | $65.0 \pm 0.4$ | $\mathbf{75.4} \pm 0.4$ | $75.8 \pm 8.6$ | $75.6 \pm 0.4$ | $76.9 \pm 1.3$ | $\mathbf{87.5} \pm \mathbf{0.8}$ |
| | 512 | $\mathbf{65.3} \pm 0.6$ | $75.2 \pm 0.7$ | $\mathbf{70.8} \pm 6.0$ | $\mathbf{76.6} \pm 1.0$ | $\mathbf{77.5} \pm 2.2$ | $87.0 \pm 0.6$ |
| CIFAR-100 | 32 | $20.3 \pm 0.1$ | $28.1 \pm 0.5$ | $462.3 \pm 15.1$ | $49.3 \pm 0.6$ | $37.9 \pm 1.6$ | $69.5 \pm 1.6$ |
| | 64 | $23.7 \pm 0.9$ | $33.4 \pm 0.5$ | $286.4 \pm 10.4$ | $58.7 \pm 0.7$ | $49.4 \pm 1.4$ | $78.7 \pm 1.6$ |
| | 128 | $28.4 \pm 0.3$ | $39.1 \pm 0.7$ | $139.1 \pm 11.8$ | $65.6 \pm 0.7$ | $64.3 \pm 2.3$ | $85.9 \pm 0.7$ |
| | 256 | $32.2 \pm 0.5$ | $45.1 \pm 0.6$ | $117.4 \pm 8.2$ | $\mathbf{66.0} \pm 0.9$ | $\mathbf{64.6} \pm 1.8$ | $\mathbf{86.4} \pm 0.5$ |
| | 512 | $\mathbf{34.4} \pm 0.5$ | $\mathbf{47.1} \pm 0.5$ | $115.3 \pm 9.0$ | $65.9 \pm 0.5$ | $64.3 \pm 1.4$ | $85.9 \pm 0.5$ |

Table 2: Overparametrization study: for a latent dimension equal to 64, we vary the width of the generator (confidence intervals computed on 10 checkpoints). Increasing the capacity of GANs tend to structure their latent space in simplicial clusters (better LogReg accuracy) and improve their performance on precision, density and coverage.

## 5 CONCLUSION

This paper aims to make a step toward a better understanding of GANs learning disconnected distributions. When the latent space dimension is large enough, we present an optimal latent space geometry of GANs: 'simplicial clusters', a Voronoi partition where each cell is a convex cone. We further show experimentally that GANs with sufficient latent capacity tend to respect this optimal geometry. We believe that our analysis can foster exciting research on GANs, with both theoretical and practical impacts. For example, understanding the optimal latent space's geometry could help design semi-supervised or transfer algorithms from GANs. Also, it could inspire new neural architectures with a bias for this 'simplicial cluster' partitioning of the latent space. Finally, let us note that our results could potentially be extended to other types of generative models with Gaussian latent space and, thus, would allow a better understanding of their properties. To adapt our analysis for variational auto-encoders or diffusion models, one would need to adapt our results to a stochastic generator. This could be an exciting follow-up of our work.

**Limitations.** We showed the existence of optimal generators and have shown experimentally that overparametrization plays a key role. However, a limitation of our work is that we could not prove their uniqueness. This is linked to partitions with the lowest $\varepsilon$-boundaries measures in the Gaussian space, which is a complex, unknown result. A second limitation is that the derived optimal generators are not valid in the case $m > d + 1$, because the minimizers of Gaussian isoperimetric inequalities are not known in this case.

**Potential negative societal impacts.** This work is mainly about understanding the behavior of deep generative models. Thus, it may lead to practical improvements in this technology and increase its potential negative impacts, such as *deepfakes* (Fallis, 2020).

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

# A  IMPLEMENTATION DETAILS

Table 3: GANs training details on MNIST

| Operation | Kernel | Strides | Feature Maps | Activation |
|---|---|---|---|---|
| Generator G(z) | | | | |
| $z \sim \mathrm{N}(0,I)$ | | | $dim(z)$ | |
| Fully Connected | | | $7 \times 7 \times 128$ | |
| Convolution | $3 \times 3$ | $1 \times 1$ | $7 \times 7 \times 64$ | LReLU |
| Convolution | $3 \times 3$ | $1 \times 1$ | $7 \times 7 \times 64$ | LReLU |
| Nearest Up Sample | | | $14 \times 14 \times 64$ | |
| Convolution | $3 \times 3$ | $1 \times 1$ | $14 \times 14 \times 32$ | LReLU |
| Convolution | $3 \times 3$ | $1 \times 1$ | $14 \times 14 \times 32$ | LReLU |
| Nearest Up Sample | | | $14 \times 14 \times 32$ | |
| Convolution | $3 \times 3$ | $1 \times 1$ | $28 \times 28 \times 16$ | LReLU |
| Convolution | $3 \times 3$ | $1 \times 1$ | $28 \times 28 \times 1$ | Tanh |
| D(x) | | | $28 \times 28 \times 1$ | |
| Convolution | $4 \times 4$ | $2 \times 2$ | $14 \times 14 \times 512$ | LReLU |
| Convolution | $3 \times 3$ | $1 \times 1$ | $14 \times 14 \times 512$ | LReLU |
| Convolution | $4 \times 4$ | $2 \times 2$ | $7 \times 7 \times 512$ | LReLU |
| Convolution | $3 \times 3$ | $1 \times 1$ | $7 \times 7 \times 512$ | LReLU |
| Fully Connected | | | 1 | - |
| Batch size | 256 | | | |
| Leaky ReLU slope | 0.2 | | | |
| Gradient Penalty weight | 10 | | | |
| Learning Rate Discriminator | $1 \times 10^{-4}$ | | | |
| Learning Rate Generator | $5 \times 10^{-5}$ | | | |
| Disciminator steps | 2 | | | |
| Optimizer | Adam | $\beta_1 : 0.5$ | $\beta_2 : 0.5$ | |

**Training.**  We use the Wasserstein loss with gradient-penalty on interpolations of fake and real data. At each iteration, the discriminator is trained 2 steps and the generator 1 step with Adam optimizer. The batch size is 256. The learning rate of the discriminator is two times larger (Heusel et al., 2017), *i.e.* $5 \times 10^{-5}$ for the generator and $1 \times 10^{-4}$ for the discriminator. GANs are trained for 80k steps on MNIST and for 100k steps on CIFAR datasets. Architectures of generator and discriminator are described in Table 3 and Table 4.

**Evaluation.**  Features are extracted with a classifier with simple architecture (convolutions, relu activation, no batch normalization). The classifier is trained on each dataset with cross-entropy loss. Weights of the classifiers are given in the code. For evaluation metrics, we follow the setting proposed by the authors. For FID (Heusel et al., 2017), we use 50k real images and 50k fake images. For precision, recall, density and coverage (Kynkäänniemi et al., 2019; Naeem et al., 2020), we use 10k real images and 10k fake images with nearest-k= 5. Full results of the study on latent space dimension are presented in Table 5.

We also share the code for a better reproducibility.

**GPUs.**  For all datasets, the training of GANs was run on NVIDIA TESLA V100 GPUs (16 GB). The training of a GAN for 100k steps on CIFAR takes around 30 hours.

Table 4: GANs training details on CIFAR datasets. BN stands for batch-normalization.

| Operation | Kernel | Strides | Feature Maps | Conditional BN (Chen et al., 2018b) | Activation |
|---|---|---|---|---|---|
| Generator G(z) | | | | | |
| $z \sim \mathrm{N}(0, Id)$ | | | 128 | | |
| Fully Connected | | | $4 \times 4 \times 128$ | - | |
| ResBlock | $[3 \times 3] \times 2$ | $1 \times 1$ | $4 \times 4 \times 128$ | Y | ReLU |
| Nearest Up Sample | | | $8 \times 8 \times 128$ | - | |
| ResBlock | $[3 \times 3] \times 2$ | $1 \times 1$ | $8 \times 8 \times 128$ | Y | ReLU |
| Nearest Up Sample | | | $16 \times 16 \times 128$ | - | |
| ResBlock | $[3 \times 3] \times 2$ | $1 \times 1$ | $16 \times 16 \times 128$ | Y | ReLU |
| Nearest Up Sample | | | $32 \times 32 \times 128$ | - | |
| Convolution | $3 \times 3$ | $1 \times 1$ | $32 \times 32 \times 3$ | - | Tanh |
| Discriminator D(x) | | | $32 \times 32 \times 3$ | | |
| ResBlock | $[3 \times 3] \times 2$ | $1 \times 1$ | $32 \times 32 \times 256$ | - | ReLU |
| AvgPool | $2 \times 2$ | $1 \times 1$ | $16 \times 16 \times 256$ | - | |
| ResBlock | $[3 \times 3] \times 2$ | $1 \times 1$ | $16 \times 16 \times 256$ | - | ReLU |
| AvgPool | $2 \times 2$ | $1 \times 1$ | $8 \times 8 \times 256$ | - | |
| ResBlock | $[3 \times 3] \times 2$ | $1 \times 1$ | $8 \times 8 \times 256$ | - | ReLU |
| ResBlock | $[3 \times 3] \times 2$ | $1 \times 1$ | $8 \times 8 \times 256$ | - | ReLU |
| Mean spatial pooling | - | - | 256 | - | |
| Fully Connected | | | 1 | - | - |
| Batch size | 256 | | | | |
| Gradient Penalty weight | 10 | | | | |
| Learning Rate Discriminator | $1 \times 10^{-4}$ | | | | |
| Learning Rate Generator | $5 \times 10^{-5}$ | | | | |
| Discriminator steps | 2 | | | | |
| Optimizer | Adam | $\beta_1 = 0.$ | $\beta_2 = 0.999$ | | |

| Dataset | L. Dim. | Acc. | FD | Prec. | Rec. | Dens. | Cov. |
|---|---|---|---|---|---|---|---|
| MNIST | 2 | $44.3 \pm 2.5$ | $112.3 \pm 22.6$ | $2.9 \pm 1.0$ | $89.1 \pm 2.0$ | $0.9 \pm 0.3$ | $2.1 \pm 0.5$ |
| | 4 | $59.4 \pm 0.7$ | $36.4 \pm 9.3$ | $75.7 \pm 1.8$ | $91.5 \pm 0.5$ | $47.9 \pm 2.2$ | $66.0 \pm 2.0$ |
| | 6 | $83.9 \pm 0.4$ | $15.4 \pm 3.0$ | $90.0 \pm 0.7$ | $92.4 \pm 0.7$ | $76.3 \pm 2.0$ | $85.8 \pm 0.4$ |
| | 8 | $84.7 \pm 0.4$ | $18.2 \pm 3.4$ | $91.1 \pm 0.7$ | $92.3 \pm 0.5$ | $81.3 \pm 2.2$ | $87.1 \pm 0.8$ |
| | 10 | $86.7 \pm 0.4$ | $10.7 \pm 2.6$ | $93.3 \pm 0.5$ | $91.5 \pm 0.5$ | $89.0 \pm 2.2$ | $89.8 \pm 0.7$ |
| | 12 | $87.4 \pm 0.4$ | $7.2 \pm 1.6$ | $94.6 \pm 0.5$ | $91.5 \pm 0.5$ | $94.3 \pm 2.0$ | $91.8 \pm 0.7$ |
| | 16 | $86.9 \pm 0.4$ | $11.0 \pm 1.0$ | $93.8 \pm 0.3$ | $91.8 \pm 0.6$ | $92.2 \pm 0.9$ | $90.7 \pm 0.5$ |
| | 32 | $90.3 \pm 0.4$ | $10.4 \pm 1.6$ | $93.6 \pm 0.3$ | $91.6 \pm 0.7$ | $91.8 \pm 1.7$ | $90.3 \pm 0.6$ |
| | 64 | $90.4 \pm 0.3$ | $13.3 \pm 2.6$ | $93.2 \pm 0.4$ | $92.3 \pm 0.5$ | $90.1 \pm 2.0$ | $90.2 \pm 0.8$ |
| | 128 | $87.0 \pm 0.4$ | $11.0 \pm 2.3$ | $93.6 \pm 0.3$ | $91.8 \pm 0.4$ | $91.6 \pm 1.5$ | $90.4 \pm 0.7$ |
| CIFAR-10 | 2 | $22.7 \pm 1.7$ | $450.5 \pm 25.6$ | $5.9 \pm 0.7$ | $83.9 \pm 1.0$ | $1.8 \pm 0.2$ | $4.1 \pm 0.4$ |
| | 5 | $35.3 \pm 0.5$ | $183.1 \pm 10.7$ | $64.0 \pm 0.6$ | $82.6 \pm 0.6$ | $51.8 \pm 1.0$ | $74.8 \pm 0.7$ |
| | 10 | $52.5 \pm 0.3$ | $117.6 \pm 7.2$ | $71.2 \pm 0.4$ | $82.3 \pm 0.5$ | $65.6 \pm 1.1$ | $82.3 \pm 0.7$ |
| | 30 | $64.2 \pm 0.5$ | $89.8 \pm 7.0$ | $75.1 \pm 0.7$ | $81.4 \pm 0.8$ | $76.0 \pm 1.8$ | $85.9 \pm 0.8$ |
| | 128 | $64.1 \pm 0.5$ | $81.3 \pm 10.7$ | $75.6 \pm 0.5$ | $81.8 \pm 0.5$ | $76.2 \pm 1.6$ | $86.7 \pm 0.6$ |
| Synthetic | 2 | $25.0 \pm 3.0$ | $175.9 \pm 19.6$ | $7.9 \pm 0.8$ | $78.3 \pm 1.2$ | $2.4 \pm 0.3$ | $5.1 \pm 0.7$ |
| CIFAR-10 | 5 | $35.3 \pm 0.5$ | $183.1 \pm 10.7$ | $64.0 \pm 0.6$ | $82.6 \pm 0.6$ | $51.8 \pm 1.0$ | $74.8 \pm 0.7$ |
| | 10 | $48.3 \pm 0.6$ | $131.7 \pm 28.5$ | $58.8 \pm 4.0$ | $78.2 \pm 1.0$ | $59.0 \pm 6.2$ | $74.6 \pm 3.7$ |
| | 30 | $57.3 \pm 0.9$ | $57.7 \pm 11.1$ | $73.6 \pm 2.0$ | $76.7 \pm 0.8$ | $79.3 \pm 5.4$ | $90.4 \pm 1.3$ |
| | 128 | $55.7 \pm 1.0$ | $79.0 \pm 12.4$ | $71.3 \pm 2.9$ | $77.7 \pm 1.6$ | $71.4 \pm 8.1$ | $84.2 \pm 6.3$ |
| CIFAR-100 | 2 | $12.1 \pm 1.2$ | $912.4 \pm 125.8$ | $1.7 \pm 0.3$ | $86.2 \pm 0.9$ | $0.5 \pm 0.1$ | $0.9 \pm 0.1$ |
| | 5 | $11.1 \pm 0.3$ | $244.9 \pm 8.7$ | $49.5 \pm 0.8$ | $81.9 \pm 0.3$ | $34.9 \pm 0.9$ | $64.4 \pm 1.2$ |
| | 10 | $21.7 \pm 0.3$ | $220.2 \pm 10.2$ | $57.7 \pm 0.7$ | $81.6 \pm 0.7$ | $47.7 \pm 1.3$ | $76.9 \pm 1.0$ |
| | 30 | $27.3 \pm 0.4$ | $198.7 \pm 14.8$ | $61.2 \pm 0.6$ | $80.7 \pm 0.6$ | $54.6 \pm 1.4$ | $81.7 \pm 0.9$ |
| | 64 | $28.4 \pm 0.3$ | $139.1 \pm 11.8$ | $65.6 \pm 0.7$ | $79.5 \pm 0.6$ | $64.3 \pm 2.3$ | $85.9 \pm 0.7$ |
| | 128 | $26.7 \pm 0.5$ | $191.9 \pm 10.7$ | $61.7 \pm 0.8$ | $80.2 \pm 0.8$ | $56.4 \pm 1.2$ | $82.5 \pm 1.0$ |

Table 5: Performance of GANs when varying latent space dimension. Confidence intervals are computed on 10 checkpoints of the same training. See main paper for curves of precision and FID with regard to the latent space dimension.

# B  TECHNICAL RESULTS

## B.1  PROOF OF LEMMA 1

We want to show that generator $G \in \mathscr{G}_L^{\mathscr{A}}$ is such that $\alpha_G \leqslant 1 - \gamma(\partial^{\varepsilon_{\min}} \mathscr{A})$, where

$$\partial^{\varepsilon_{\min}} \mathscr{A} = \bigcup_{i=1}^{m} \left( \cup_{j \neq i} A_j \right)^{\varepsilon_{\min}} \setminus \left( \cup_{j \neq i} A_j \right),$$

*Proof by contradiction.*

Assume a generator $G$ such that there exists $z \in \partial^{\varepsilon_{\min}} \mathscr{A}$ and $i \in [1, m]$ such that $G(z) \in M_i$. Since $G$ is associated with $\mathscr{A}$, we have using Definition 2, that there exists $z'$ and $j \in [1, m], j \neq i$ such that $\|z - z'\| < \varepsilon_{\min}/2$ and $j = \arg\min_{k \in [1,m]} \|G(z') - M_k\|$. Thus, we have:

$$\|G(z) - G(z')\| \geqslant d(G(z'), M_i),$$
$$\geqslant d(M_i, M_i)/2,$$
$$\geqslant D_{\min}/2.$$
$$\text{And, } \frac{\|G(z) - G(z')\|}{\|z - z'\|} > D_{\min}/\varepsilon_{\min},$$
$$> L.$$

This contradicts $G$ being in $\mathscr{G}_L^{\mathscr{A}}$.

## B.2  PROOF OF THEOREM 1

**Proving that: for $m \leqslant d + 1$, for any $\delta > 0$, if $L$ is large enough, then, for any well-balanced $G \in \mathscr{G}_L$, we have $\alpha_G \leqslant 1 - \gamma(\partial^{\varepsilon_{\min}} \mathscr{A}^\star) + \delta$.** Let $G$ be a well-balanced generator and $\mathscr{A}$ the partition associated with $G$. Let us first define the gaussian boundary measure $P_\gamma$ of a partition $\mathscr{A}$ of $\mathbb{R}^d$. For partitions with smooth boundaries, it coincides with the $(d-1)$-dimensional gaussian measure of the boundary, defined as follows:

$$P_\gamma(\mathscr{A}) = \liminf_{\varepsilon \to 0} \frac{\gamma(\partial^\varepsilon \mathscr{A}) - \gamma(\mathscr{A})}{\sqrt{2/\pi}\varepsilon}$$

Moreover, for sets with smooth boundaries, we have from Federer (1969, Theorem 3.2.29):

$$\liminf_{\varepsilon \to 0} \frac{\gamma(\partial^\varepsilon \mathscr{A}) - \gamma(\mathscr{A})}{\sqrt{2/\pi}\varepsilon} = \lim_{\varepsilon \to 0} \frac{\gamma(\partial^\varepsilon \mathscr{A}) - \gamma(\mathscr{A})}{\sqrt{2/\pi}\varepsilon}$$

Let us denote $\mathscr{A}^\star$, the optimal partition defined in Milman and Neeman (2022), based on simplicial clusters. $A^\star$ is a standard partition where $\gamma(A_1^\star) = \ldots = \gamma(A_m^\star)$ for all i, and $\sum_i \gamma(A_i) = 1$. By the multi-bubble theorem (Milman and Neeman, 2022), simplicial clusters (such as $\mathscr{A}^\star$) are the unique minimizers of the gaussian isoperimetric problem, thus:

$$P_\gamma(\mathscr{A}^\star) \leqslant P_\gamma(\mathscr{A})$$
$$\lim_{\varepsilon \to 0} \frac{\gamma(\partial^\varepsilon \mathscr{A}^\star)}{\varepsilon} \leqslant \lim_{\varepsilon \to 0} \frac{\gamma(\partial^\varepsilon \mathscr{A})}{\varepsilon}$$
$$L_{\mathscr{A}} \leqslant L_{\mathscr{A}^\star}$$

where $L_{\mathscr{A}} = \lim_{\varepsilon \to 0} \frac{\gamma(\partial^\varepsilon \mathscr{A}^\star)}{\varepsilon}$ and $L_{\mathscr{A}^\star} = \lim_{\varepsilon \to 0} \frac{\gamma(\partial^\varepsilon \mathscr{A}^\star)}{\varepsilon}$.

Then, for any $\delta > 0$, there exists $\varepsilon' > 0$ such that, for any $\varepsilon < \varepsilon'$,

$$\left| \frac{\gamma(\partial^\varepsilon \mathscr{A}^\star)}{\varepsilon} - L_{\mathscr{A}^\star} \right| < \delta \quad , \quad \left| \frac{\gamma(\partial^\varepsilon \mathscr{A})}{\varepsilon} - L_{\mathscr{A}} \right| < \delta \quad \text{and} \quad L_{\mathscr{A}^\star} \leqslant L_{\mathscr{A}}$$

Thus, for any $\delta > 0$, there exists $\varepsilon' > 0$ such that, for any $\varepsilon < \varepsilon'$,

$$\gamma(\partial^\varepsilon \mathscr{A}^\star) \leqslant \gamma(\partial^\varepsilon \mathscr{A}) + 2\delta\varepsilon \tag{7}$$

Besides, we know that

$$\alpha_G \leqslant 1 - \gamma(\partial^{\varepsilon_{\min}} \mathscr{A})$$

Consequently, we have that:

$$\alpha_G \leqslant 1 - \gamma(\partial^{\varepsilon_{\min}} \mathscr{A})$$
$$\leqslant 1 - \gamma(\partial^{\varepsilon_{\min}} \mathscr{A}^\star) + 2\delta\varepsilon_{\min} \quad \text{using (7).}$$

We conclude by choosing $L$ big enough such that $\varepsilon_{\min}$ is strictly smaller than $\frac{\delta}{\delta/2}$.

**Second part of Theorem 1.** Let $L, D$ be such that $L \geqslant D\sqrt{\log(m)}$. Let's prove that for any well-balanced generator $G \in \mathscr{G}_L$, we have:

$$\alpha_G \leqslant 1 - \varepsilon_{\min}\sqrt{\log m}\, e^{-3/2}.$$

Using the method from Schechtman (2012), we have the measure of the border of cell $i$:

$$\gamma\Big(\big(\cup_{j\neq i} A_j\big)^\varepsilon \setminus \big(\cup_{j\neq i} A_j\big)\Big) \geqslant \frac{1}{\sqrt{2\pi}} \int_t^{t+\varepsilon} e^{-s^2/2} ds, \quad \text{where } t \text{ is such that } \frac{1}{\sqrt{2\pi}} \int_t^\infty e^{-s^2/2} ds = 1/m,$$

$$\geqslant \frac{\varepsilon}{\sqrt{2\pi}} e^{-(t+\varepsilon)^2/2},$$

$$\geqslant \frac{\varepsilon\sqrt{\log m}}{m} e^{-\varepsilon t - \varepsilon^2/2} \quad (\text{using } \sqrt{\log m} \leq t \leq \sqrt{2\log m}),$$

$$\geqslant \frac{\varepsilon\sqrt{\log m}}{m} e^{-\varepsilon\sqrt{\log m} - \varepsilon^2/2}.$$

Thus:

$$\gamma(\partial^{\varepsilon_{\min}} \mathscr{A}) = \sum_{i=1}^m \gamma\Big(\big(\cup_{j\neq i} A_j\big)^\varepsilon \setminus \big(\cup_{j\neq i} A_j\big)\Big) \geqslant \varepsilon_{\min}\sqrt{\log m}\, e^{-\varepsilon_{\min}\sqrt{\log m} - \varepsilon_{\min}^2/2}.$$

Thus, we have

$$\alpha_G \leqslant 1 - \gamma(\partial^{\varepsilon_{\min}} \mathscr{A}),$$
$$\leqslant 1 - \varepsilon_{\min}\sqrt{\log m}\, e^{-\varepsilon_{\min}\sqrt{\log m} - \varepsilon_{\min}^2/2}.$$

Moreover, using $\varepsilon_{\min} = \frac{D}{L}$ and $L \geqslant D\sqrt{\log m}$, so we get $\varepsilon_{\min}\sqrt{\log m} \leqslant 1$:

$$\alpha_G \leqslant 1 - \varepsilon_{\min}\sqrt{\log m}\, e^{-3/2}.$$

### B.3 PROOF OF THEOREM 2

For a given partition $\mathscr{A}$, and a target distribution $\mu^\star$ with m disconnected components $M_i, i \in [1, m]$, we defined $X_i, i \in [1, m]$ a set of sampled data points such that for all $i \in [1, m]$, we have $X_i \in M_i$. Now, we define $G_\varepsilon^\star$ with $\varepsilon > 0$, a generative model such that:

$$G_\varepsilon^\star(z) = \sum_{i \in S_z} w_i(z)\, X_i, \quad \text{with } w_i(z) = \frac{d(z, (A_i^\varepsilon)^\complement)}{\sum_{j \in S_z} d(z, (A_j^\varepsilon)^\complement)} \tag{8}$$

where $d(z, A) = \min_{a \in A} \|z - a\|$, and $S_z = \{j \in [1, n] \text{ such that } z \in A_j^\varepsilon\}$ denotes the set of cell-extensions the point $z$ belongs to. We can see that $G_\varepsilon^\star \sharp \gamma$ memorizes the dataset since every $z$ close to the center of the cell $A_i$ such that $|S_z| = 1$ verifies $G_\varepsilon^\star(z) = X_i$. An illustration is given in Figure 2.

To be more precise, all samples are mapped to one of the center of the modes $X_i, i \in [1, m]$, except for those in $\varepsilon$-boundaries. When z belongs to the intersection of two $\varepsilon$-boundaries, $G_\varepsilon(z)$ is a simple linear combination of 2 points. It is only when $|S_z| \geqslant 3$ that more complex samples are generated. A simple illustration of $G_\varepsilon^\star$ for $d = 2$ and $m = 3$ is given in Figure 2. Interestingly, one can also show that the image of $G_\varepsilon^\star$ is equal to the convex hull of the diracs $X_i, i \in [1, m]$. In particular, there exists a particularly interesting neighborhood $v$ of 0 where $G_\varepsilon^\star(v)$ is equal to the whole convex hull of the points $X_i, i \in [1, m]$.

**Proof that $G^\star_\varepsilon$ is well-balanced.** We recall that a generator is *well-balanced* if we have $G\sharp\gamma(M_1) = \ldots = G\sharp\gamma(M_m)$. By construction (8), we have that for any $i \in [1,m]$

$$\|G^\star_\varepsilon(z) - X_i\| = \|\sum_{k\neq i} w_k(X_k - X_i)\|,$$
$$= D \times (1 - w_i).$$

So, for any $z \in A_i$, we have that

$$i = \underset{j\in[1,m]}{\arg\min} \; w_j = \underset{j\in[1,m]}{\arg\min} \; \|G(z) - X_j\|$$

Thus $G^\star_\varepsilon$ is associated with the optimal partition $\mathscr{A}^\star,$ .

Besides, for a given radius $r$ of the different modes, since everything is symmetrical, we have that $\gamma(\{z \in \mathbb{R}^d, \|G(z) - X_1\| \leqslant r\}) = \ldots = \gamma(\{z \in \mathbb{R}^d, \|G(z) - X_m\| \leqslant r\})$. Thus, the generator is well-balanced.

**Proof that $G^\star_{\varepsilon_{\max}}$ is in $\mathscr{G}_L$.** To begin with, we have $\mathbb{R}^d = \partial^\varepsilon\mathscr{A} \cup (\mathbb{R}^d \setminus \partial^\varepsilon\mathscr{A})$.

$$\mathbb{R}^d \setminus \partial^\varepsilon\mathscr{A} = \bigcup_{i=1}^m A_i \setminus ((\cup_{j\neq i}A_j)^\varepsilon \cap A_i),$$

and, we know that for each $i \in [1,m]$, $G^\star_\varepsilon(z)$ is constant and thus $L$-Lipschitz.

Now,

$$\partial^\varepsilon\mathscr{A} = \bigcup_{i=1}^m (\cup_{j\neq i}A_j)^\varepsilon \setminus (\cup_{j\neq i}A_j) = \bigcup_{\substack{S\in\mathscr{P}([1,m]) \\ \mathrm{card}(S)\geqslant 2}} \bigcap_{i\in S} A_i^\varepsilon.$$

Now, let $S \in \mathscr{P}([1,m])$ with $\mathrm{card}(S) = k \geqslant 2$. Let $z,z' \in S^2$. Let $\alpha = (\alpha_1,\ldots,\alpha_m)$ and $\beta = (\beta_1,\ldots,\beta_m)$ be two vectors, both in $\mathbb{R}^m$, such that for all $i \in [1,m]$:

$$\alpha_i = \frac{d(z,(A_i^\varepsilon)^\complement)}{\sum_{j\in\mathscr{A}_z} d(z,(A_j^\varepsilon)^\complement)} \quad \text{and} \quad \beta_i = \frac{d(z',(A_i^\varepsilon)^\complement)}{\sum_{j\in\mathscr{A}_z} d(z',(A_j^\varepsilon)^\complement)}$$

We have that

$$\|G(z) - G(z')\| = \|(1 - \sum_{i\neq 1}\alpha_i)X_1 - (1 - \sum_{i\neq 1}\beta_i)X_1 + \sum_{i\neq 1}\alpha_i X_i - \sum_{i\neq 1}\beta_i X_i\|$$
$$= \|\sum_{i\neq 1}(\alpha_i - \beta_i)(X_1 - X_i)\|$$
$$\leqslant \max_{(i,j)\in[1,m]^2} \|X_i - X_j\| \; \|\alpha - \beta\|,$$
$$\leqslant \max_{(i,j)\in[1,m]^2} \|X_i - X_j\| \; \|h(z) - h(z')\|,$$

where $h$ is the function from $\mathbb{R}^d \to \mathbb{R}^m$ defined as:

$$h(z) = \left(\frac{d(z,(A_1^\varepsilon)^\complement)}{\sum_{i\in\mathscr{A}_z} d(z,(A_i^\varepsilon)^\complement)},\ldots,\frac{d(z,(A_m^\varepsilon)^\complement)}{\sum_{i\in\mathscr{A}_z} d(z,(A_i^\varepsilon)^\complement)}\right).$$

We can write $h = f \circ g$ with $f$ the function defined from $\mathbb{R}^d \to \mathbb{R}^m$ by

$$f(z) = \left(d(z,(A_1^\varepsilon)^\complement),\ldots,d(z,(A_k^\varepsilon)^\complement)\right),$$

and $g$ the function defined on $\mathbb{R}^m \setminus \{0\}$ by

$$g(z) = \frac{z}{\|z\|_1}$$

We have that $f$ is a $\sqrt{m}$-Lipschitz functions (given that $z \mapsto d(z, (A_m^{\varepsilon})^{\complement})$ is 1-Lipschitz). Besides, we know that outside the ball $B_{\varepsilon}(0)$, the function $g$ is $(1/\varepsilon)$-Lipschitz.

Using the convexity of function $z \mapsto \sum_{j \in \mathscr{A}_z} d(z, (A_j^{\varepsilon}))$ (as a sum of convex functions), we can show that for all $z \in \mathscr{A}_z$, we have that $f(z) \leqslant (m-1)\varepsilon$ and $f(z)$ is not $B_{\varepsilon}(0)$. Finally, the function $h$ is $\frac{\sqrt{m}}{\varepsilon}$-Lipschitz.

Thus, we have that:

$$\|G_{\varepsilon}^{\star}(z) - G_{\varepsilon}^{\star}(z')\| \leqslant \frac{D\sqrt{m}}{\varepsilon}\|z - z'\|,$$

with $D = \|X_i - X_j\|, (i,j) \in [1,m]^2, i \neq j$. Consequently, by noting $\varepsilon_{\max} = \frac{D\sqrt{m}}{L}$, we have :

$$\|G_{\varepsilon_{\max}}^{\star}(z) - G_{\varepsilon_{\max}}^{\star}(z')\| \leqslant L\|z - z'\|.$$

We can now conclude on the Lipschitzness of $G^{\star}$ on $\mathbb{R}^d$.

**Proving that: for $m \leqslant d+1$, for any $\delta > 0$, if $L$ is large enough, then, for any well-balanced $G \in \mathscr{G}_L$, we have $\alpha_{G_{\varepsilon_{\max}}^{\star}} \geqslant \alpha_G - \delta$.** From the proof of Theorem 1, we have that for any $\delta > 0$, there exists $\varepsilon_{\min}$ such that:

$$\begin{aligned}
\alpha_G &\leqslant 1 - \gamma(\partial^{\varepsilon_{\min}}\mathscr{A}) \\
&\leqslant 1 - \gamma(\partial^{\varepsilon_{\min}}\mathscr{A}^{\star}) + 2\delta\varepsilon_{\min} \quad \text{using (7)}.
\end{aligned}$$

Now, by construction of $G_{\varepsilon_{\max}}^{\star}$, we have that

$$\alpha_{G_{\varepsilon_{\max}}^{\star}} \geqslant 1 - \gamma(\partial^{\varepsilon_{\max}}\mathscr{A}^{\star}).$$

Consequently,

$$\begin{aligned}
\alpha_G &\leqslant 1 - \gamma(\partial^{\varepsilon_{\min}}\mathscr{A}^{\star}) + 2\delta\varepsilon_{\max} + \gamma(\partial^{\varepsilon_{\max}}\mathscr{A}^{\star}) - \gamma(\partial^{\varepsilon_{\max}}\mathscr{A}^{\star}) \\
&\leqslant \alpha_{G_{\varepsilon}^{\star}} + 2\delta\varepsilon_{\max} + \gamma(\partial^{\varepsilon_{\max}}\mathscr{A}^{\star}) - \gamma(\partial^{\varepsilon_{\min}}\mathscr{A}^{\star}) \\
&\leqslant \alpha_{G_{\varepsilon}^{\star}} + 2\delta\varepsilon_{\max} + \gamma(\partial^{\varepsilon_{\max}}\mathscr{A}^{\star}) - 2L_{\mathscr{A}^{\star}}\varepsilon_{\max} - \gamma(\partial^{\varepsilon_{\min}}\mathscr{A}^{\star}) + 2L_{\mathscr{A}^{\star}}\varepsilon_{\min} + 2L_{\mathscr{A}^{\star}}(\varepsilon_{\max} - \varepsilon_{\min}) \\
&\leqslant \alpha_{G_{\varepsilon}^{\star}} + 4\delta\varepsilon_{\max} + 2L_{\mathscr{A}^{\star}}\varepsilon_{\max}, \\
&\leqslant \alpha_{G_{\varepsilon}^{\star}} + \varepsilon_{\max}(4\delta + 2L_{\mathscr{A}^{\star}}).
\end{aligned}$$

We conclude by choosing $L$ big enough such that $\varepsilon_{\max}$ is strictly smaller than $\frac{\delta}{4\delta + 2L_{\mathscr{A}^{\star}}}$.

**Proving the second part of Theorem 2.** The precision of $G_{\varepsilon}^{\star}$ is thus such that:

$$\alpha_{G_{\varepsilon_{\max}}^{\star}} \geqslant 1 - \gamma(\partial^{\varepsilon_{\max}}\mathscr{A}).$$

However, since $\partial^{\varepsilon}\mathscr{A} \subset \bigcup_{i=1}^n A_i^{\varepsilon}$, we have that for any $\varepsilon$

$$\gamma(\partial^{\varepsilon}\mathscr{A}) \leqslant \sum_{i=1}^n \gamma(A_i^{\varepsilon}).$$

Using results from Schechtman (2012, Proposition 1), when $m \leq d$, there exists $C$ large enough, such that

$$\gamma(A_i^{\varepsilon_{\max}}) \leqslant \frac{\varepsilon_{\max}}{m}\left(\sqrt{\pi \log(Cm)}\right).$$

Thus, we have

$$\alpha_{G_{\varepsilon_{\max}}^{\star}} \geqslant 1 - \varepsilon_{\max}\sqrt{\pi \log(Cm)},$$

To have $\alpha_{G_{\varepsilon_{\max}}^{\star}} \geqslant 0$, we must have $\varepsilon_{\max} \leqslant 1/\sqrt{\pi \log(Cm)}$. This is the case since we have

$$\varepsilon_{\max} = D\sqrt{m}/L \quad \text{and} \quad L \geqslant D\sqrt{m}\sqrt{\pi \log(Cm)}.$$

## C  ADDITIONAL RESULT

**Impact of the number of modes.**   To illustrate the results from Theorem 1 and Theorem 2, we propose to vary the number of modes of the data distribution. On real-world data, the number of modes is set but usually unknown, and removing/adding classes as a proxy for modes usually does not give insightful results since some classes can be much more complex than others. We thus use a synthetic setting, where we can easily control both the number of modes and their complexity. Figure 7 stresses that as the number of modes increase, the precision decrease. Interestingly, using large latent space dimension can relieve the problem, even if the latent space dimension is clearly below that of the target. Recall the two problems that arise when training GANs: i) *dimensional misspecification* where the true and modeled distributions do not have density functions w.r.t. the same base measure, and ii) *density misspecification*, where GANs try to fit a disconnected manifold with a unimodal disitribution. From the results we conclude that:

- With very low latent space dimensions, both problems i) and ii) have to be addressed and this leads to poor precision as the number of modes increases.
- With larger latent space dimensions, the problem i) is less of a burden even when there is a clear dimensional misspecification and thus the GANs' performance is more tied to problem ii).

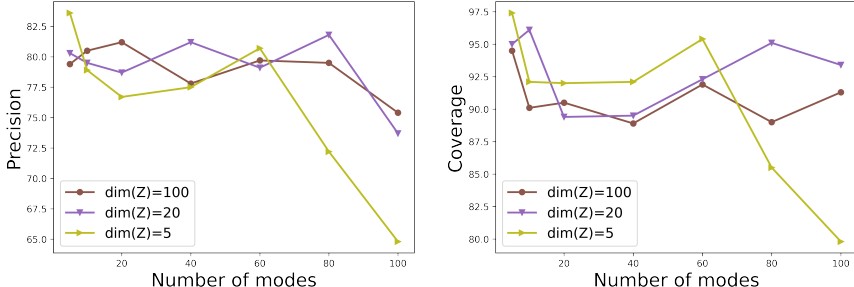

Figure 7: Training on a mixture of Gausians in $\mathbb{R}^{100}$ with varying number of modes and varying latent space dimension. The bigger the number of modes, the lower the precision. Increasing the latent space dimension helps up to a limit depending on the number of modes.

