# OpenReview forum: "On the optimal precision of GANs"
_ICLR.cc/2023/Conference — Submitted to ICLR 2023_

### Official Review · Reviewer_q2EK · 2022-10-21

**Confidence:** 4
**Correctness:** 2
**Technical Novelty And Significance:** 2
**Empirical Novelty And Significance:** 2
**Recommendation:** 3

**Clarity, Quality, Novelty And Reproducibility:**

Clarity:

- In Theorem 1, what is $\delta$?
- The definition of the Density and Coverage needs more clarification. Besides, in Table 2, how do you compute density and coverage?


**Strength And Weaknesses:**

Strengths:

The paper poses an interesting angel towards understanding generative models (not restricted to GANs). A Lipschitz generator of cannot map a unimodal distribution to disconnected distributions. Therefore, there is a tradeoff between mode-collapse and precision of the generator. Through Gaussian Isoperimetric inequality, the simplicity and regularity of the latent space (linearity & convexity) can be related to the quality of the generation.


Weaknesses:

Theory:

1. The assumptions are too strong and fail to capture the essence of the real generative models.

(1) The theory depends on the $L_2$ Lipschitz continuity $L$ of the generator network. For deep models $L$ is often a very large constant. Just think about the existence of adversarial examples.

(2) The abstraction of the target distribution is artificial. The paper assumes the target distribution is a mixture of $m$ balls of equal distance. This assumption can be readily relaxed without modifying the main theorems, e.g., just defind $D$ as the minimum distance between those modes.

2. The technical contributions of the theorems are weak.

(1) The only interesting theoretical result is Lemma 1, which bridges the surface area of the latent space partition and the precision of the generator. The result is weak since empirically $D/L$ is very small and hence the upper bound is loose. Then the behavior of the generator is influenced by other factors.

(2) Theorem 1 & 2 are straight applications of the Gaussian Isoperimetric inequality.

Experiments:

1. The experiments are based on an implicit assumption that different classes correspond to different disconnected "modes". Besides, the empirical results (LogReg Acc & Convex Acc) are weak, suggesting latent space partition according to different classes is not predicted by the theory.

2. Several experiments are disconnected from the theory. E.g., latent space dimension, disentangling the manifold dimension, and the Impact of overparametrization. Those experiments are far-fetched from the theory part.


**Summary Of The Paper:**

A Lipschitz-continuous generator of GAN cannot map a unimodal distribution to disconnected distributions. Therefore, in order to transform a unimodal distribution, e.g., Gaussian, of the latent variable to cover all modes of the disconnected target distribution, the generator must map a subset of the latent space outside the support of the target distribution.

In order to minimize the probability of the generator outputting an out-of-support sample, the paper studies some sufficient condition of the shape of the latent space partitions via Isoperimetric inequality. By invoking results from (Milman and Neeman, 2022), the author indicates that the latent space forms a  Voronoi partition when $ m < d+1$ ($m$ is the number of modes, $d$ is the latent space dimension). Next, the authors perform several experiments on structure of the latent space of GANs, e.g., the linear separability of the latent space partition, the convexity fo the latent space partition, etc.

**Summary Of The Review:**


The paper bridges the configuration of the latent space of a generator and the precision through Isoperimetric inequality. The idea is interesting but there are several weaknesses in both the theory and the experiments.

---

> ### Author Response · Authors · 2022-11-08
> **Answer to reviewer q2EK**
>
> We want to thank you for the time dedicated to the review.
>
> * The theory depends on the  Lipschitz continuity  of the generator network. For deep models L is often a very large constant. Just think about the existence of adversarial examples.
>
> Yes, we agree that deep models with huge capacity have large Lipschitz constants L. Actually, this remark validates the setting of our study that require large Lipschitz constants. In this case, the $\epsilon$-boundary is small and, thus, we approach the setting of isoperimetric inequalities from [3].
>
> * The abstraction of the target distribution is artificial. This assumption can be readily relaxed without modifying the main theorems, e.g., just define D as the minimum distance between those modes.
>
> Thank you for your suggestion. We had thought of relaxing this assumption, but the constraint of the equal distance is a bit more complex than that. Our theory requires the generator to be associated with a simplicial cluster in the latent space. This issue is discussed in the subsection "What if modes are not equally distant?" on page 6 of the paper. If points are aligned for example, it is not be possible to define a Lipschitz generator associated with a simplicial cluster partition. Assumption 1 has the benefits of being clear and simple, and still leaves room for interesting insights.
>
> * In Theorem 1, what is $\delta$?
>
> The use of $\delta$ in Theorem 1 has a technical purpose. Since we can pick $\delta$ infinitely close to 0, the partition associated to a simplicial cluster is approximately optimal. This $\delta$ is used in the same way in Theorem 2.
>
> * The empirical results (LogReg Acc and Convex Acc) are weak suggesting latent space partition according to different classes is not predicted by the theory.
>
> We would like to stress that, on the contrary, our experiments tend to show that the latent space partition obtained in practice is close to the one predicted by the theory. The models with the best precision have the highest LogReg Acc in Table 1 and Table 2. In the  overparametrization study (Table 2), the Pearson correlation between precision and LogReg Acc is extremely high: $0.99$ for CIFAR 10 and $0.91$ for CIFAR100. This supports the link between the performance of GANs and our theoretical results. Similarly, other works on ImageNet (BigBiGAN [1]) have showed that the latent space of GANs reaches 60.1\% top-1 accuracy and 81.9\% top-5 accuracy on the latent linear probing test.
>
> * Several experiments are disconnected from the theory. E.g., latent space dimension, disentangling the manifold dimension, and the Impact of overparametrization.
>
> We argue that these experiments are not disconnected from the theory, but rather help validating the theoretical analysis. The experiment "Latent space dimension" proves that when the latent dimension is greater than a threshold, the precision does not increase anymore: this is in par with our theoretical analysis (our bounds only require $d > m+1$). "Disentangling the manifold dimension" validates that with low latent dimension, the precision drops because of the multi-modality of the data. Besides, "Overparametrization" shows that, when the capacity of the generator increases, both precision and linear separability improve. This also validates our theory.
>
> * How do you compute density and coverage?
>
> We compute it as proposed in the original paper [2], with $N=10000$ points, $k=5$ for the nearest neighbors, and same feature extractor than the other metrics. The package has been open-sourced by the authors: https://pypi.org/project/prdc/.
>
> [1] Donahue, J., \& Simonyan, K. (2019). Large scale adversarial representation learning. Advances in neural information processing systems, 32.
>
> [2] Naeem, M. F., Oh, S. J., Uh, Y., Choi, Y., \& Yoo, J. (2020, November). Reliable fidelity and diversity metrics for generative models. In International Conference on Machine Learning (pp. 7176-7185). PMLR.
>
> [3] Milman, E., & Neeman, J. (2022). The Gaussian double-bubble and multi-bubble conjectures. Annals of Mathematics, 195(1), 89-206.

---

### Official Review · Reviewer_ApLT · 2022-10-25

**Confidence:** 4
**Correctness:** 3
**Technical Novelty And Significance:** 3
**Empirical Novelty And Significance:** 1
**Recommendation:** 5

**Clarity, Quality, Novelty And Reproducibility:**

The paper becomes clearer as it progresses. Though there are some issues with the clarity of the definitions, the paper is overall clear. The results are novel and rely on a novel new GI characterization. No further issues along these lines.

**Strength And Weaknesses:**

Strengths:

The paper adopts a novel perspective on the latent space of GANs, by using results in Gaussian isoperimetry (GI) to conjecture about the form of the learned space of optimally-learned GANs. Using this perspective and powerful tools from GI, the authors try to relate precision to the structure of the latent space learned by a GAN. This is a really nice idea. The proofs are fairly clearly and straightforwardly done, with an exception (see below).


Weaknesses:

The overall structure of the theory is not supported by the practical considerations.

The experiments don't really support the theoretical results as much as they could. The theory consists of an upper bound on precision which holds for any model, and a lower bound which holds for an "optimal" model. The lower bound is inherently more interesting in this situation, but its statement is weaker. A natural next question is: do the backprop-learned GANs actually follow this lower bound?

The relevant quantities are L, D, and m, which can all be measured to empirically investigate this question. The theorems state a number of tradeoffs betweeen these quantities, which could be empirically investigated without much effort, but are not. The lower/upper bounds don't actually match, because the quantification over the partitions is different; and I'd therefore like to see empirical verification of the \sqrt{log m} rate. This is easy to test not only for MNIST, but more multiclass problems like ImageNet-1000; you can subsample classes and get precision numbers.

The issue of precision itself is a little muddled in this paper. It's not clearly defined initially. Then it's assumed that r << D, i.e. effectively that the data are on a set of m point masses. This ignores the data-dependent effects of distance with space, and leads to the fundamental D/L scaling of the epsilons in the theorems. How to interpret the theorems on real data is therefore somewhat unclear.

The Lipschitz constant L is a parametrization the authors choose, which ends up making the proofs track with proofs of the analogous GI results. Like other alternative parametrizations, it is something that could be reasonably estimated in practice - what is it for the networks used in the experiments? Is L large enough in practice to support the desired bounds? From the cited Virmaux+Scaman reference, L is ~150 for an MNIST CNN of depth 5; it would be good to include the results of such an evaluation, to complement these experiments. Meanwhile, D can also be estimated for the dataset, and m is known. So the bounds on precision and recall can actually be numerically computed and compared. This would provide some notion of how good the parametrization is, and how much more theoretical understanding is needed.

It would also be good to see the "convex accuracy" reported for all the experiments - I find that and the logistic regression to be very flexible and useful aids to understanding the relationship between simplicial clusters and the empirical learned space. Experimenting on more networks, grid-searching across approximate {L, D, m} configurations, is needed.

Relaxing the GI assumption to encompass data-dependent Gaussian concentration of measure (without isoperimetric symmetries) would be a very interesting avenue of future work.

**Summary Of The Paper:**

This paper uses a recent mathematical result -- about the form of Gaussian-isoperimetric partitionings of Euclidean space -- to conjecture that successfully learned GANs learn such "optimal" partitionings of the space. This is used to try to quantify the precision of such learned GANs in terms of other parameters of the problem. The paper contains experiments on GAN latent spaces, on which the data are Gaussian by construction, and which can link learned GAN latent spaces to theory.

**Summary Of The Review:**

The overall idea, of using deep results in Gaussian isoperimetry to explore learned GAN representations, is great and could be a very useful avenue of study in several ways. The theoretical narrative is very incomplete, and though the experiments are promising, the theory is not sufficiently supported or explored by the experiments.

---

> ### Author Response · Authors · 2022-11-08
> **Answer to reviewer ApLT**
>
> First, we want to thank you for the time you dedicated to the review and for your interesting remarks on our paper.
>
> * The experiments don't really support the theoretical results as much as they could. [...] The relevant quantities are L, D, and m, which can all be measured to empirically investigate this question. The theorems state a number of tradeoffs betweeen these quantities, which could be empirically investigated without much effort, but are not. [...] Experimenting on more networks, grid-searching across approximate {L, D, m} configurations, is needed.
>
> Thank you for raising this interesting point. However, there are several difficulties for running such experiments.
>
> 1) Difficulty of estimating L. The algorithm from Virmaux and Scaman only estimates an upper-bound of L, which they say is loose for large networks. For example, for AlexNet, the estimated upper-bound is $3.62\times10^7$.
> Moreover, since L depends on the weights of the trained generator, it is difficult to do a grid-search over L.
> 2) Difficulty of estimating m. On ImageNet, for example, there are some very diverse classes that have several modes. The number of classes in the dataset is not necessarily equal to the number of modes in the image space, which would make the curve noisy.
> 3) We have done a study on synthetic Mixture of Gaussians (end of the appendix), and have shown that when the number of modes increase (all modes have equidistant means), there is a slow decrease in terms of precision, but the curve is noisy. Computing this curve on a real dataset with many classes such as ImageNet would be very costly from a computational viewpoint.
>
>
>
>
> * The issue of precision itself is a little muddled in this paper. It's not clearly defined initially. Then it's assumed that r << D, i.e. effectively that the data are on a set of m point masses. This ignores the data-dependent effects of distance with space [...].
>
> We would like to note that we use in the current paper, the precision as it was defined in [2] (Definition 1). Also, the assumption r<<D only impacts the distances in between two classes with respect to distances within one class. The distance D in between two classes (data-dependent) is still very important in our analysis.
>
> * It would also be good to see the "convex accuracy" reported for all the experiments.
>
> Thank you for this suggestion, the results will be soon added to the new version of the paper.
>
> * Relaxing the GI assumption to encompass data-dependent Gaussian concentration of measure (without isoperimetric symmetries) would be a very interesting avenue of future work.
>
> We agree that Assumption 1 is a strong assumption. Interestingly, Theorem 1 from [2] shows that simplicial clusters are minimizers of the gaussian isoperimetric inequality even if the partition is unequal. Thus, the optimality results of the simplicial cluster in Theorem 1 and 2 would also hold, but not the bound since those were computed for a zero-centered simplicial cluster. We will state this more clearly in the paper.

---

### Official Review · Reviewer_jLeB · 2022-10-28

**Confidence:** 3
**Clarity, Quality, Novelty And Reproducibility:** See above.
**Correctness:** 3
**Technical Novelty And Significance:** 3
**Empirical Novelty And Significance:** 2
**Recommendation:** 6

**Strength And Weaknesses:**

The usage of the theoretical result seems valid. However, it still seems that the assumption (Assumption 1) is a bit too strong. In practice, the prior on different modes can be unbalanced. Some discussion is provided but no solution is proposed. Also the distribution can be a mixture model instead of the case that each distribution is strictly restricted within a (d+1)-dimensional Voronoi cell.

The implication of the bound is unclear. No practical insights are revealed as a consequence of the bound. The experiments explore the geometry like linearly separability and convexity. But these results are still quite far from the bound.

**Summary Of The Paper:**

This paper uses existing theoretical results on the boundary measure of dimensional Voronoi cells of m<d+1 equal distance modes to show a bound of the precision of a m-mode GAN. Some empirical study is carried out to study the geometry of the latent space. However, the empirical result is not particularly well-connected with the theoretical bound.

**Summary Of The Review:**

Interesting usage of an existing theory for GAN precision bounds. But the theoretical results is not well connected with practical usage of GANs.

---

> ### Author Response · Authors · 2022-11-08
> **Answer to reviewer jLeB**
>
> We want to thank you for the time you dedicated to the review and for your interesting remarks on our paper.
>
> * The experiments explore the geometry like linearly separability and convexity. But these results are still quite far from the bound.
>
> We would like to note that one of the main result of our paper is to stress that generators with latent space structured as a simplicial cluster have optimal precision levels. It is only thanks to this result that we have derived the lower-bound in Theorem 2. Therefore, our experiments aim at verifying if GANs have the geometric properties of simplicial clusters (linear separability and convexity).
>
> Besides, knowing that GANs have these properties is interesting in itself, and might explain certain observed phenomenons such as 1) successful editing operations made from the latent space, or 2) why the latent space of GANs is good for representation learning: BigBiGAN's latent space reaches 81.9\% top-5 accuracy on the linear probing test [1].
>
> * It still seems that the assumption (Assumption 1) is a bit too strong. In practice, the prior on different modes can be unbalanced. Some discussion is provided but no solution is proposed.
>
> Thank you for your remark.  Interestingly, Theorem 1 from [2] shows that simplicial clusters are minimizers of the gaussian isoperimetric inequality even if the partition is unequal. Besides, to get a simplicial cluster with different mixture weights, you can just translate the center from the origin. Thus, the optimality results of the simplicial cluster would hold. However, the bounds would not hold, since it comes from the measure of $\epsilon$-boundaries of zero-centered simplicial clusters.
>
>
> * The implication of the bound is unclear. No practical insights are revealed as a consequence of the bound.
>
> First, it shows that, if your latent space dimension is large enough, there is a slow decrease of precision with regards to the number of modes. It also shows that networks with larger Lipschitz constant L (attainable with large models) can reach higher precision. It supports the empirical practice to use large latent space dimension, as shown in Figure 6 of our paper.
>
> [1] Donahue, J., \& Simonyan, K. (2019). Large scale adversarial representation learning. Advances in neural information processing systems, 32.
>
> [2] Milman, E., \& Neeman, J. (2022). The Gaussian double-bubble and multi-bubble conjectures. Annals of Mathematics, 195(1), 89-206.

---

> > ### Comment · Reviewer_jLeB · 2022-12-03
> > **Post Rebuttal Response**
> >
> > I appreciate the authors clarified some of my misunderstandings. I still think the theorem is not sufficiently helpful to the real world applications. But I do think it would be a good theoretical contribution to the community. I am raising my score to 6.

---

### Official Review · Reviewer_xjYQ · 2022-10-29

**Confidence:** 4
**Correctness:** 3
**Technical Novelty And Significance:** 3
**Empirical Novelty And Significance:** 3
**Recommendation:** 6

**Clarity, Quality, Novelty And Reproducibility:**

The paper’s contributions are novel and carried out with sufficient quality and clarity, except for some minor issues I discussed above. Sufficient details (as well as code) makes the paper very reproducible.

**Strength And Weaknesses:**

**Strengths**

The paper is well-written and easy to follow. The posed question is also very interesting to me, understanding how partitioning of the latent space affects the precision of GANs, and vice versa, finds many potential applications for improving GAN precision and searching/editing in their latent space. The Theorems and claims are mostly correct, and the provided bounds are non-trivial and interesting.


**Weaknesses**

In general I like the paper, but I have some questions and minor concerns that I’ll go over below:

1) The paper neither shows Voronoi partitioning to be a necessary nor a sufficient condition for achieving optimal precision. While the authors do touch on this point in several parts of the paper (including right after Theorem 2 and then again at limitations), I still think the language is vague and must be made very explicit to avoid misleading a not so very careful reader. In particular, since the paper starts by raising the following question in its abstract: “what is the latent space partition that minimizes the measure of out-of-manifold samples”, a reader is likely to think that the answer is a Voronoi partition, yet this is not true! The question remains open in that no necessary condition on the latent space is discovered. Please make this explicit in your abstract – mentioning the question is great for motivation, but it needs to follow the explicit mention that you could not answer this question, rather could find one particular generator with Voronoi partitioning that minimizes the measure of out-of-manifold samples, which is still plenty valuable as a starting point in my opinion.

2) In Figure 1’s caption, can you be more specific about in what sense left is close to right?

3) The equal distance assumption is not well justified, either cite or elaborate more specifically.

4) Can you be more specific on the statement “If L is large enough”; and elaborate in what way small L breaks the theory, and discuss any connections to practical settings where L cannot be very large without causing instabilities.

5) The claim in Table 1 that “the higher the precision is, the more each cell in the latent space is linearly separable” is not strictly true when comparing Precision and LogReg columns, same goes for the Convex column. Moreover, the table lacks error bars, so significance of any trend is questionable. Including a proper correlation test can help clarify the extent of your claim.


**Summary Of The Paper:**

This paper studies the optimal precision of GANs trained on disconnected distributions under some simplifying assumptions about the distributions. The paper derives lower and upper bounds for the precision of GANs under these assumptions, and also studies what partitioning of the latent space that can give rise to the optimal precision in GANs. The paper also provides some smaller scale experiments on image datasets that show GANs tend to construct the predicted Voronoi partitioning to a certain extent, and also behave consistently with the predicted effects of the dimensionality of the latent space on precision.


**Summary Of The Review:**

Overall, I like the paper and think the findings about what kind of latent space partitioning can be optimal for GANs with respect to precision are valuable. The provided bounds are also insightful for designing new GANs. My main concern is that the paper does not restrict its claims as well as I’d consider appropriate.

### Update post-rebuttal
I thank the authors for their response and answering my concerns. I think the paper merits acceptance upon making its assumptions and restrictions more clear, but I agree with the other reviewers that the assumptions are somewhat strong, as such keeping my score at 6.

---

> ### Author Response · Authors · 2022-11-08
> **Answer to reviewer xjYQ**
>
> We want to thank you for the time you dedicated to the review and for your interesting suggestions which will improve our paper.
>
> * The paper neither shows Voronoi partitioning to be a necessary nor a sufficient condition for achieving optimal precision.  While the authors do touch on this point in several parts of the paper (including right after Theorem 2 and then again at limitations), I still think the language is vague and must be made very explicit to avoid misleading a not so very careful reader. [...] Please make this explicit in your abstract.
>
> Thank you for this remark. We will make this explicit in our abstract to avoid any confusion.
>
> * In Figure 1’s caption, can you be more specific about in what sense left is close to right?
>
> We have updated the caption. What we meant was that, on the right, the latent space partition is very similar to an intersection between the propeller (appearing on the left) and the unit sphere.
>
> * The equal distance assumption is not well justified, either cite or elaborate more specifically.
>
> Thank you for the suggestion. We have updated the paper and cite appropriate work on concentration of distance in high-dimensional spaces.
>
>  The equal-distance assumption is necessary to construct a generator associated with a simplicial cluster.
>  We analyse this issue in the section "What if modes are not equally distant?" on page 6 of our paper. If points are aligned, it is not be possible to define a Lipschitz generator associated with a simplicial cluster.
>
>
> * Can you be more specific on the statement “If L is large enough”; and elaborate in what way small L breaks the theory, and discuss any connections to practical settings where L cannot be very large without causing instabilities.
>
> As you pointed, our theoretical analysis is based on isoperimetric inequality minimizers, and is only valid for large L (function of $\varepsilon$) where boundaries in the latent space have low measures.
> For small and fixed L, we can not say anything about the structure of minimizers of the $\epsilon-$boundary measures, since there are no mathematical results - as far as we know. The Theorem 2 from [1] motivates the use of bigger architectures with large Lipschitz constants since it was shown that small Lipschitz constants lead to poor precision.
>
> * The claim in Table 1 that “the higher the precision is, the more each cell in the latent space is linearly separable” is not strictly true when comparing Precision and LogReg columns, same goes for the Convex column. Moreover, the table lacks error bars, so significance of any trend is questionable. Including a proper correlation test can help clarify the extent of your claim.
>
> Indeed, there are other factors of variation that impact the precision of a generator. However, in the  overparametrization study (Table 2),  we have a fixed dataset and architecture, and only change the width. The Pearson correlation between precision and LogReg Acc is extremely high: $0.99$ for CIFAR 10 and $0.91$ for CIFAR100. This supports the link between the performance of GANs and linear separability.
>
> [1] Tanielian, U., Issenhuth, T., Dohmatob, E., \& Mary, J. (2020, November). Learning disconnected manifolds: a no gan’s land. In International Conference on Machine Learning (pp. 9418-9427). PMLR.

---

### Official Review · Reviewer_GvAm · 2022-11-04

**Confidence:** 2
**Correctness:** 4
**Technical Novelty And Significance:** 3
**Empirical Novelty And Significance:** 3
**Recommendation:** 6

**Clarity, Quality, Novelty And Reproducibility:**

- Suppose that, rather than Gaussian latents, we use the other common choice of latents being uniform over a hypercube (or perhaps a sphere, or a ball). It seems that isoperimetry is probably easier to study in this case; are there parallel results that can be immediately adapted to this setting? (Do GANs follow those results?)

- The question "what if nodes are not equally distant?" seems quite important. Are cats, dogs, frogs, and trucks really equally distant? What's even the right notion of distance here, for a given generator architecture – surely it's not Euclidean distance in pixel space, as even for Lipschitz generators there'll be some warping of the space, but it also can't really be something based on the particular generator parameters since the question is about what those parameters will become? Is there anything we can say about the formal setting here? Anything we can look at in the experiments for specifically this kind of question? (If you, say, download a pretrained unconditional ImageNet GAN, do the latent space divisions have some relationship to distance in the WordNet hierarchy between classes, or between word embeddings?)

Overall, I found the paper easy to read, the assumptions more or less reasonable, and the theoretical results clearly explained. (I did not check the proofs, and am not particularly familiar with work on isoperimetry beyond basic definitions; nor have I really kept up with the GAN literature for the past few years.)

**Strength And Weaknesses:**

Strengths:
- The paper gives a clear exposition of a fairly thorough examination of the results in question.
- It increases our understanding of "optimal" latent space organization for Gaussian latents.

Weaknesses:
- The scope of the results is somewhat limited.
- A few interesting questions are raised without being fully explored.

**Summary Of The Paper:**

This paper adapts recent results on the Gaussian isoperimetric inequality to study the latent space organization of GANs with disconnected modes. This proves bounds on the precision of GANs in certain circumstances, and implies various properties about how these latent spaces "should" be organized.

**Summary Of The Review:**

An interesting paper bringing recent mathematical results to a relevant community who probably wouldn't otherwise hear about them, while also exploring GAN-specific properties in an interesting way. Although the paper has its limitations, I think it's a nice step in the understanding of latent-space generative models.

---

> ### Author Response · Authors · 2022-11-08
> **Answer to reviewer GvAm**
>
> We want to thank you for the time you dedicated to the review and for your interesting remarks on our paper.
>
> * What about other latent distributions?
>
> Indeed, a similar reasoning is applicable for any latent distribution where the isoperimetric inequality results are known. Interestingly, in a very recent paper [1], the authors show that optimal partitions in Euclidean or Spherical spaces have properties similar to simplicial clusters. We chose the specific Gaussian case since it is the one often used when training GANs.
>
> * On the notion of "equal-distance" between modes. Are cats, dogs, frogs, and trucks really equally distant? What is even the right notion of distance here?
>
> The question of distance between modes is indeed important. We mainly argue that the "equal-distance" between modes can be justified with arguments from high-dimensional space analysis [3,4] where each mode ("dog" or "cat") would be an independent sample from a high-dimensional distribution.
>
> The notion of distance is also very interesting but intricate. The euclidean distance can be disconnected from a semantic one. More plausible is that the distance between the generator's inner-representations of the different modes might play an important role. A recent paper [5] has showed that in the context of classification, embeddings of different classes tend to converge around their means, all of which are equidistant from one another and structured in equiangular simplex.
>
> * Is there anything we can say about the formal setting here?
>
>  The equal-distance assumption is necessary to construct a generator associated with a simplicial cluster.
>  We analyse this issue in the section "What if modes are not equally distant?" on page 6 of our paper. If points are aligned for example, it is not be possible to define a Lipschitz generator associated with a simplicial cluster partition.
>
> * If you, say, download a pretrained unconditional ImageNet GAN, do the latent space divisions have some relationship to distance in the WordNet hierarchy between classes, or between word embeddings?
>
> This is actually a really interesting question. Does the distance in the latent space correlate with a semantic distance given by the WordNet hierarchy. This would be interesting for future research. Note that authors from [2] have trained a large-scale unconditional GAN on ImageNet and obtained 81.9\% top-5 accuracy on the linear probing test. This shows that the latent space of GANs are well linearly separated.
>
>
> [1] Milman, E., \& Neeman, J. (2022). The Structure of Isoperimetric Bubbles on $\mathbb {R}^ n $ and $\mathbb {S}^ n$. arXiv preprint arXiv:2205.09102.
>
> [2] Donahue, J., \& Simonyan, K. (2019). Large scale adversarial representation learning. Advances in neural information processing systems, 32.
>
> [3]  Beyer, K., Goldstein, J., Ramakrishnan, R., and Shaft, U.. When is “nearest neighbor”
> meaningful? In International conference on database theory, pages 217–235. Springer, 1999.
>
> [4] Aggarwal, C. C., Hinneburg, A., \& Keim, D. A. (2001, January). On the surprising behavior of distance metrics in high dimensional space. In International conference on database theory (pp. 420-434). Springer, Berlin, Heidelberg.
>
> [5] Kothapalli, V., Rasromani, E., & Awatramani, V. (2022). Neural Collapse: A Review on Modelling Principles and Generalization. arXiv preprint arXiv:2206.04041.

---

> > ### Comment · Reviewer_GvAm · 2022-11-29
> > **Belated response**
> >
> > Thanks for your prompt comments here (and sorry that I drafted a response to this weeks ago but apparently never hit submit!).
> >
> > I think you’ve helped clarify the contributions here. I’m going to maintain my score, though, since I think the assumptions are just kind of strong for the paper to be informative about practical models. I’d feel more positive about a version of this paper with more in-depth investigation into the related behaviour of real models or relaxation of the assumptions for the analysis to at least approximately apply.

---

### Decision · Program_Chairs · 2023-01-20

**Decision:**

Reject

**Justification For Why Not Higher Score:**

The overall formulation of the theory is now well-supported by the practical aspects of GANs.

**Justification For Why Not Lower Score:**

N/A

**Metareview: Summary, Strengths And Weaknesses:**

The work sheds light onto the key aspects of GANs, from mode collapse to their precision/recall performance using the perspective of Gaussian isoperimetry inequalities, focusing on the case of disconnected distributions. The writing, while vague at times (esp. on the Voronoi partitioning's role for achieving optimal precision), is easy to follow. The insights are somewhat in line with what we expect though the theory provides more like guidance than a tight characterization.

The reviewers helped improve the paper, clarifying the assumptions as well as some of the definitions.

Unfortunately, the relevance to the real data setting is still unclear, blocking the impact of the paper. While the authors made improvements on the text during the discussion period, there needs to be further revisions.